

# Ruling out light axions: The writing is on the wall

## Konstantin A. Beyer$^\star$ and Subir Sarkar

Rudolf Peierls Centre for Theoretical Physics, Parks Road, Oxford OX1 3PU, UK

$\star$ konstantin.beyer@mpi-hd.mpg.de

## Abstract

We revisit the domain wall problem for QCD axion models with more than one quark charged under the Peccei-Quinn symmetry. Symmetry breaking during or after inflation results in the formation of a domain wall network which would cause cosmic catastrophe if it comes to dominate the Universe. The network may be made unstable by invoking a 'tilt' in the axion potential due to Planck scale suppressed non-renormalisable operators. Alternatively the random walk of the axion field during inflation can generate a 'bias' favouring one of the degenerate vacua, but we find that this mechanism is in practice irrelevant. Consideration of the axion abundance generated by the decay of the wall network then requires the Peccei-Quinn scale to be rather low — thus e.g. ruling out the DFSZ axion with mass below $\sim 11$ meV, where most experimental searches are in fact focused.

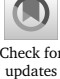

# 1   Introduction

Despite the many successes of the Standard Model (SM) of particle physics a number of important questions remain unanswered. For example stable SM matter (i.e. nucleons) accounts for only $\sim 5\%$ of the total energy density of the universe, while $\sim 26\%$ is in the form of dark matter (DM) [1]. While DM is likely constituted of new relic, weakly interacting particles, no experiment has yet detected its non-gravitational interactions hence its fundamental nature remains elusive.

An attractive candidate particle for DM motivated *within* the SM is the axion. This is the pseudo Nambu-Goldstone boson arising from the spontaneous symmetry breaking of a chiral $U(1)_{\mathrm{PQ}}$ introduced by Peccei & Quinn (PQ) [2,3] to solve the 'strong-$CP$' problem, viz. why do strong interactions not violate charge-parity symmetry, thereby generating an electric dipole moment (EDM) of the neutron which is not observed [4]. When the 'Weinberg-Wilczek' axion corresponding to such symmetry breaking at the electroweak scale [5,6] was not found, it was realised that the Peccei-Quinn scale $f_{\mathrm{PQ}}$ can be much higher, implying an 'invisible axion' with very suppressed couplings to SM fields. Nevertheless such relic axions can account for the cold dark matter of the universe for $f_{\mathrm{PQ}} \sim 10^{9-11}$ GeV [7–9] as coherent oscillations of the axion field have the same equation of state as non-relativistic particles.

The cosmological evolution is even more interesting because of a sequence of symmetry breaking which produces potentially stable topological defects. Below $f_{\mathrm{PQ}}$ the vacuum manifold is not simply connected. This implies the existence of closed paths in physical space which get mapped onto non-trivial paths in field space winding around the origin. Such field configurations correspond to cosmic strings [10–12]. When the temperature drops to be of $\mathcal{O}(\Lambda_{\mathrm{QCD}}) \sim 300$ MeV, QCD instantons generate a mass for the axion [13,14]:

$$m_a^2(T(t)) = 1.7 \times 10^{-7} \frac{\Lambda_{\mathrm{QCD}}^4}{f_{\mathrm{PQ}}^2} \left( \frac{\Lambda_{\mathrm{QCD}}}{T} \right)^{6.7} \Rightarrow m_a(T=0) \simeq 5.7 \, \mu\mathrm{eV} \left( \frac{10^{12} \, \mathrm{GeV}}{f_{\mathrm{PQ}}} \right). \quad (1)$$

This breaks the symmetry to $Z(N_{\mathrm{DW}})$, where $N_{\mathrm{DW}}$ is the number of quarks charged under $U(1)_{\mathrm{PQ}}$ [15,16]. The vacuum manifold of $Z(N_{\mathrm{DW}})$ is however disconnected which implies the existence of paths in physical space which map onto paths interpolating between two vacuum states in field space. Such paths necessarily leave the vacuum manifold and the resulting structure is a domain wall (DW).

Topologically each string must be connected by $N_{\mathrm{DW}}$ domain walls once the axion gets a mass. It has been argued that due to the surface energy of domain walls, a network of strings and domain walls with $N_{\mathrm{DW}} = 1$ would be unstable and collapse [17]. Hadronic axion models like KSVZ [18,19] have $N_{\mathrm{DW}} = 1$, however, most models have $N_{\mathrm{DW}} > 1$, e.g. the DFSZ axion [20,21] has $N_{\mathrm{DW}} = 6$. Wall networks with $N_{\mathrm{DW}} > 1$ are stable and can lead to cosmological catastrophe if they come to dominate the energy density of the universe after they form [15], which is inevitable because of the slower scaling of their energy density than that of radiation or matter. This happens at a time [13]:

$$t_{\mathrm{dom}} \lesssim \frac{3}{32\pi G_{\mathrm{N}} \sigma_{\mathrm{DW}}} \simeq 53 \, \mathrm{s} \left( \frac{10^{12} \, \mathrm{GeV}}{f_{\mathrm{PQ}}} \right), \quad (2)$$

where the wall tension is $\sigma_{\mathrm{DW}} \simeq 9 m_a f_{\mathrm{PQ}}^2 \simeq 5.1 \times 10^{10} \, \mathrm{GeV}^3 (f_{\mathrm{PQ}}/10^{12} \, \mathrm{GeV})$. This marks the latest time by which the walls must have decayed, else the universe becomes dominated by them and enters a stage of accelerated (power-law) expansion with no end [22]. In fact a much stronger bound is obtained by considering the effects of the wall decay products.

Axions emitted by the decaying string-domain wall network must be accounted for in calculating the total relic abundance of axions, along with those from the standard misalignment

mechanism. This provides a prediction of the mass for axions to constitute the dark matter, thereby sharpening the relevant target space for experimental searches. Most investigations of the post-inflation PQ scenario, including the contributions from axion strings, indicate the range $m_a \sim 10^{-5} - 10^{-3}$ eV [23,24] — the 'light axion' window. This is therefore where most experimental searches are focussed, especially those using tunable microwave cavities [25,26]. This does not however take into account the potential contribution from domain walls. Moreover such axions are born relativistic with a non-thermal spectrum, and turn non-relativistic subsequently. This is quite different from both the 'cold' axions from the misalignment mechanism which have the equation of state of a non-relativistic gas, and any 'hot' axions created in thermal equilibrium in the early universe with a relativistic Bose-Einstein spectrum. The effect of the latter two populations on the formation of structure in the universe has been investigated in detail (for a review, see [27]). However the effect of the initially relativistic but non-thermally produced axions from domain wall decay warrants further investigation.

Lattice simulations of the axion field evolution taking into account the temperature dependence of the mass and the domain wall contribution for $N_{\mathrm{DW}} = 1$ models have recently been performed [28]. However because of the large separation of scales between the thickness ($\sim m_a^{-1}$) and the separation ($\sim H^{-1}$) of the walls, such simulations of meta-stable DW networks are challenging. (Here $H \equiv \dot{R}/R$ is the Hubble expansion rate, where $R(t)$ is the scale-factor of the universe.) We show below that axion models with $N_{\mathrm{DW}} > 1$ are severely constrained without need for such studies. Henceforth we assume for simplicity $N_{\mathrm{DW}} = 2$ which yields a 'frustrated' stable wall network; our conclusions hold also for $N_{\mathrm{DW}} > 2$ in particular the DFSZ axion which has $N_{\mathrm{DW}} = 6$.

The usual argument for evading the domain wall problem is to invoke non-renormalisable Plank-scale suppressed operators reflecting the effects of quantum gravity on global symmetries. These explicitly break the $U(1)_{\mathrm{PQ}}$ [29–31] and lift the degeneracy of the vacuum states, resulting in a pressure term which causes the true vacuum domain to grow [15] and the DW network to collapse. The co-moving domain wall energy density decays as [32]

$$\rho_{\mathrm{DW}} \propto \frac{\sigma_{\mathrm{DW}}}{\eta} \exp\left[-\mu^3 \left(\frac{\eta}{\eta_{\mathrm{DW}}}\right)^3\right], \tag{3}$$

where $\mu$ is the fractional energy difference between the potential minima, $\eta = \int \mathrm{d}t/R(t)$ is the conformal time and $\eta_{\mathrm{DW}}$ is its value when the walls form. Such explicit breaking of $U(1)_{\mathrm{PQ}}$ is experimentally constrained as it reintroduces the $CP$ violation which is required by the upper limit on the neutron EDM [33,34] to be negligibly small. Requiring axion domain walls to disappear in time to avoid cosmological catastrophe thus implies a *lower* bound on the neutron EDM. Consequently the tilt solution to the domain wall problem is falsifiable by improved experiments.

We also consider an alternative mechanism to render domain walls unstable by introducing a statistical *bias* in the population of the vacuum states. Such a bias leads, for $Z(2)$ models, to exponential decay of the co-moving domain wall energy density [32,35–37]:

$$\rho_{\mathrm{DW}} \propto \frac{\sigma_{\mathrm{DW}}}{\eta} \exp\left[-\varepsilon^2 \left(\frac{\eta}{\eta_{\mathrm{DW}}}\right)^3\right], \tag{4}$$

with $\varepsilon$ the bias and $\eta_{\mathrm{DW}}$ the conformal time when the walls form. This mechanism was proposed as a generic solution to the domain wall problem for weakly coupled fields [38]. Such bias may be generated by the dynamics of the axion field during inflation (although this has been recently disputed taking super-horizon correlations into account [39]). The random walk of the axion field during an extended inflationary epoch has been exploited to open up previously excluded axion parameter space [40,41].

This paper is organised as follows. We begin by reviewing (§ 2) the two solutions above to the domain wall problem and highlight the challenges. In particular the bias solution turns out to be irrelevant. Although the tilt solution does work, in § 3 we show that the overproduction of axions from the collapsing string-wall network effectively excludes it for most of the parameter space that axion dark matter searches are presently focussed on.[1] Experiments looking for heavier QCD axions [43–46] thus receive further motivation from this work. However the relevant parameter space for hadronic axion models e.g. KSVZ [47] is unaffected as no domain walls then survive from the early universe. Our arguments do not apply when Peccei-Quinn symmetry breaking occurs before the onset of inflation.

## 2 Solutions to the domain wall problem

### 2.1 Tilt

Proposed by Sikivie [15], the standard solution to the domain wall problem lifts the topological protection of the domain walls by introducing a tilt in the potential. Therefore only one true vacuum state remains and bubbles of false vacuum eventually collapse under the pressure stemming from the increased volume energy density within the bubbles of false vacuum. The explicit breaking of the global symmetry is due to quantum gravity effects at the scale $M_{\mathrm{QG}}$ parameterised by non-renormalisable operators [29–31]:

$$\delta V_{M_{\mathrm{QG}}} = \frac{|g|\,e^{i\delta}}{M_{\mathrm{QG}}^{2m+n-4}}\,|\phi|^{2m}\,\phi^n + \text{h.c.} + c\,, \tag{5}$$

where the constant $c$ is chosen to have $\min V = 0$. The coupling can in general be complex introducing a phase $\delta$ with coupling strength $|g|$. The above term stems from a $2m+n$-dim operator with a $U(1)_{\mathrm{PQ}}$ charge $n$; under $U(1)_{\mathrm{PQ}}$ the $|\phi|^{2m}$ stays invariant and $\phi^n$ changes by $n$. The operator is suppressed by $M_{\mathrm{QG}}^{2m+n-4}$ and we make the most conservative choice that $M_{\mathrm{QG}}$ is the Planck Scale $M_{\mathrm{Pl}} \equiv G_{\mathrm{N}}^{-1/2} \simeq 1.2 \times 10^{19}\,\text{GeV}$. If it were lower, e.g. at the string scale, this would only strengthen the bounds quoted in this paper.

After $U(1)_{\mathrm{PQ}}$ spontaneously breaks and the complex PQ field acquires a vev $v_a = N_{\mathrm{DW}}f_{\mathrm{PQ}}$, the potential (5) can be written as

$$\delta V_{M_{\mathrm{Pl}}} = |g|\,M_{\mathrm{Pl}}^2 \left(\frac{f_{\mathrm{PQ}}}{\sqrt{2}M_{\mathrm{Pl}}}\right)^{2m+n-2} f_{\mathrm{PQ}}^2 \left(1 - \cos(na+\delta)\right), \tag{6}$$

yielding a potential for the axion field below the QCD scale:

$$V(a) = m_a^2 f_{\mathrm{PQ}}^2 \left[(1 - \cos(N_{\mathrm{DW}}a)) + \mu(1 - \cos(na+\delta))\right], \tag{7}$$

with

$$\mu \equiv |g| \left(\frac{M_{\mathrm{Pl}}}{m_a}\right)^2 \left(\frac{f_{\mathrm{PQ}}}{\sqrt{2}M_{\mathrm{Pl}}}\right)^{2m+n-2}. \tag{8}$$

If $\delta$, $|g|$, $n$ and $N_{\mathrm{DW}}$ are not fine-tuned such that the potentials align perfectly, there will be only one true vacuum state.

Such additional operators are however constrained since explicit breaking of $U(1)_{\mathrm{PQ}}$ reintroduces the original strong-$CP$ problem [15]. This 'axion quality problem' is quantified as

---

[1]A similar argument has been made earlier [42] but the constraint we quote is significantly stronger.

below. The vev of the new potential (which corresponds to the QCD 'theta parameter') is:

$$\langle\theta\rangle = \frac{|g|M_{\text{Pl}}^2\left(\frac{f_{\text{PQ}}}{\sqrt{2}M_{\text{Pl}}}\right)^{2m+n-2}\frac{n}{N_{\text{DW}}}\sin\delta}{m_a^2 + |g|M_{\text{Pl}}^2\left(\frac{f_{\text{PQ}}}{\sqrt{2}M_{\text{Pl}}}\right)^{2m+n-2}n^2\cos\delta} \simeq |g|\left(\frac{M_{\text{Pl}}}{m_a}\right)^2\left(\frac{f_{\text{PQ}}}{\sqrt{2}M_{\text{Pl}}}\right)^{2m+n-2}\frac{n}{N_{\text{DW}}}\sin\delta\,, \quad (9)$$

where we have used the fact that the potential generated by the non-renormalisable operators is much smaller than the QCD potential. It is also natural to assume $\delta \sim \mathcal{O}(1)$. Requiring that the above vev respect the conservative bound $\langle\theta\rangle < 10^{-10}$ set by the experimental upper limit on the neutron EDM, we get:

$$|g|\left(\frac{f_{\text{PQ}}}{\sqrt{2}M_{\text{Pl}}}\right)^{2m+n}\frac{n}{N_{\text{DW}}} < 1.6\times10^{-91}\,, \quad (10)$$

taking $|g|$ too be of $\mathcal{O}(1)$ and using relation(1) between $m_a$ and $f_{\text{PQ}}$ for the QCD axion.

A lower bound on the tilt comes from requiring it to solve the domain wall problem. After the appearance of the domain wall network, it takes only a short while before its energy density comes to dominate the universe. The explicit breaking from the potential (6) must be large enough for the network to collapse before this happens. A conservative upper bound on the collapse time is the epoch of Big Bang nucleosynthesis (BBN) at $t_{\text{BBN}} \sim 1\,\text{s}$ [48]. Having a small explicit breaking lifts the degeneracy of the $N_{\text{DW}}$ vacuum states leaving only one true vacuum and $N_{\text{DW}} - 1$ false ones. A bubble of false vacuum surrounded by the true vacuum experiences a pressure due to the difference in energy density, causing the bubble to shrink. The domain wall contributes

$$E_{\text{DW}} = \sigma_{\text{DW}}\mathcal{R}^2\,, \quad (11)$$

for a bubble of size $\mathcal{R}$, while the energy density of the contained volume is

$$E_{\text{vol}} = \delta V\mathcal{R}^3\,. \quad (12)$$

The work done by the energy difference between the volume of false vacuum and the true vacuum at $V = 0$ is $\Delta E \sim \delta V\mathcal{R}^3$. Then we may find the force acting on the wall, $F_{\text{DW}} = \delta V\mathcal{R}^2$ and hence its acceleration:

$$|\vec{a}| = \frac{\delta V}{\sigma_{\text{DW}}} \simeq 2.8\times10^{58}\,\text{GeV}\,|g|\left(\frac{f_{\text{PQ}}}{\sqrt{2}M_{\text{Pl}}}\right)^{2m+n-1}\,, \quad (13)$$

where we estimated the potential difference $\delta V$ to be the maximum of the potential generated by non-renormalisable operators. The true difference will be slightly smaller but this does not greatly affect our argument. We can estimate the time of collapse to be roughly when the acceleration is high enough for the wall to have a velocity close to the speed of light, thereby overcoming the expansion of space and leading to collapse. This leads to the requirement:

$$|g|\left(\frac{f_{\text{PQ}}}{\sqrt{2}M_{\text{Pl}}}\right)^{2m+n-1} > 1.2\times10^{-83}\,. \quad (14)$$

The two inequalities (10) and (14) result in a constraint on the Planck scale suppressed, non-renormalisable operators required to solve the axion DW problem, while leaving the PQ solution to the strong-$CP$ problem unspoilt. Such a theory must obey

$$8.5\times10^{-91}\left(\frac{f_{\text{PQ}}}{10^{12}\,\text{GeV}}\right) < |g|\left(\frac{f_{\text{PQ}}}{\sqrt{2}M_{\text{Pl}}}\right)^{2m+n} < 1.6\times10^{-91}\frac{N_{\text{DW}}}{n}\,, \quad (15)$$

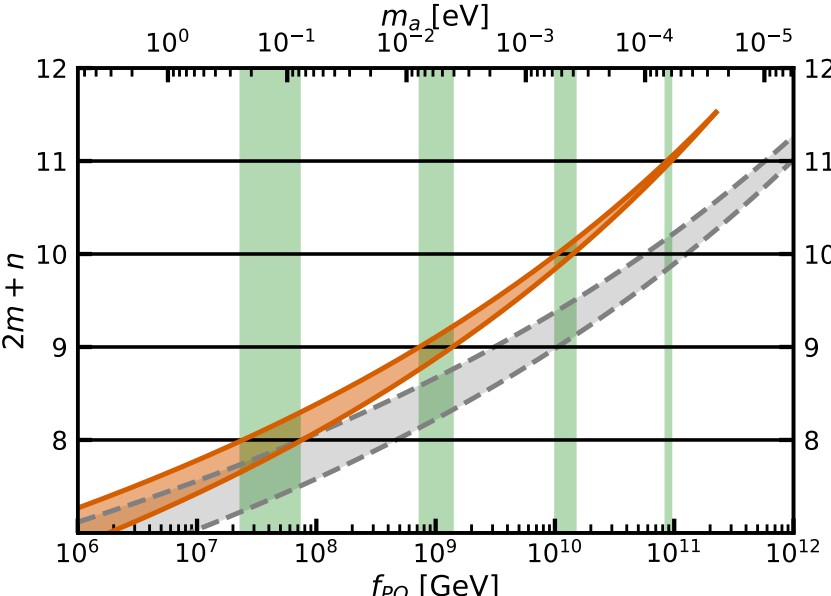

Figure 1: The dimension $2m + n$ of Planck scale suppressed non-renormalisable operators required to solve the axion domain wall problem, versus the Peccei-Quinn scale $f_{\mathrm{PQ}}$. The orange shaded region is allowed by the inequalities (17) derived in the text. The plot ends at $f_{\mathrm{PQ}} \sim 10^{11} (N_{\mathrm{DW}}/n)$ GeV above which there is no solution. Moreover $2m + n$ must be an integer — which further restricts $f_{\mathrm{PQ}}$ to the green vertical bands. The grey shaded region (bounded by dashed lines) illustrates the mild relaxation of the constraint when the coupling $|g|$ and phase $\delta$ of the tilt operator are both fine-tuned to be $10^{-2}$.

which implies:

$$f_{\mathrm{PQ}} < 1.9 \times 10^{11} \, \mathrm{GeV} \frac{N_{\mathrm{DW}}}{n} \,. \tag{16}$$

The operator dimension must thus be bounded as:

$$\frac{\log\left(1.6 \times 10^{-91} N_{\mathrm{DW}}/n\right) - \log\left(|g|\right)}{\log\left(f_{\mathrm{PQ}}/\sqrt{2}M_{\mathrm{Pl}}\right)} < 2m + n < 1 + \frac{\log\left(1.2 \times 10^{-83}\right) - \log\left(|g|\right)}{\log\left(f_{\mathrm{PQ}}/\sqrt{2}M_{\mathrm{Pl}}\right)} \,. \tag{17}$$

Figure 1 shows that whereas there do exist operators for which both the aforementioned problems are solved, the solution is very unnatural. To leave the PQ solution to the strong-$CP$ problem unspoiled we must suppress lower order operators. However, in order to have fast enough domain wall decay we must guarantee that the lowest dimensional operator which is allowed within the PQ solution does exist. We are therefore tasked with having to explain how to suppress all Planck scale suppressed operators up to the specific one we require. We conclude that explicitly breaking $U(1)_{\mathrm{PQ}}$ to get around the DW problem is inherently unsatisfactory; improvement in neutron EDM measurements will tighten the constraints further, eventually closing off this possibility altogether.

It should also be noted that $2m + n$ is an integer number which reduces the available range for $f_{\mathrm{PQ}}$ even further, e.g. $2m + n = 9$ requires $f_{\mathrm{PQ}} \sim 10^9$ GeV for tilt to solve the DW problem while not spoiling the solution to the strong-$CP$ problem.

## 2.2 Bias

Another possibility to solve the DW problem is to introduce a statistical *bias* in the distribution of the axion field. Generating an overpopulation in one of the $N_{\mathrm{DW}}$ vacuum states eventually

leads to domination by it. When the patches of the other vacua become causally connected, the tension of the domain walls makes them collapse. This was demonstrated for a $Z(2)$ symmetric potential [32, 35–37].[2]

Any field lighter than the Hubble parameter in an inflationary deSitter background, i.e. with $m_a \ll H_{\text{infl}}$, experiences quantum fluctuations of $\mathcal{O}(H_{\text{infl}})$. These fluctuations are caused by the exit of modes from the inflationary event horizon [49, 50]; each mode gives the field averaged over super-horizon scales a kick of order $H_{\text{infl}}$, while the potential causes the field to settle into the vacuum states. The interplay between these two effects is critical for a bias to appear and therefore this mechanism works only for fields which experience a potential *during* inflation. For details on the generation of a bias, see Appendix A.

The issue with this solution to the axion DW problem lies in the huge separation of the two relevant scales. The size of the field space of the axion is set by its vev $f_{\text{PQ}}$, while the step-size of the random walk the field undergoes during inflation is $\sim H_{\text{infl}}$. This leads to three distinct possibilities:

- $H_{\text{infl}} \gg f_{\text{PQ}}$: In this case, the deSitter temperature $T_{\text{deS}} \propto H_{\text{infl}}$ is higher than the symmetry breaking scale of the $U(1)_{\text{PQ}}$ and quantum fluctuations move the field back to the origin of the potential. Thus the quantum fluctuations prevent the axion field from becoming classical and no bias is generated.

- $H_{\text{infl}} \sim f_{\text{PQ}}$: As long as $H_{\text{infl}}$ is smaller than $f_{\text{PQ}}$, the PQ symmetry is broken during inflation. However, this is supposedly excluded by constraints on isocurvature perturbations [13, 51]. Additionally, the PQ scale is much higher than the QCD scale, which means that the axion does not acquire a potential from QCD instantons. Since a potential is necessary for the generation of a statistical bias, this parameter space is unsuitable for solving the domain wall problem via bias.

- $H_{\text{infl}} \ll f_{\text{PQ}}$: The hierarchy between the scales allows the PQ symmetry to be broken and the axion to develop a potential during inflation. The random walk step-size is however small and the steady state of the distribution is reached after $\sim (f_{\text{PQ}}/H_{\text{infl}})^2$ e-folds, however the mean value of the axion field has still not changed much from its initial value. To generate a bias we must wait until the mean value accumulates around the vacuum states; this can take a long time during which the causally connected patches at the time of PQ symmetry breaking inflate and any domain walls that form are exponentially large [52]. Hence this is the same as pre-inflationary PQ symmetry breaking.

To circumvent the problem with an inflationary Hubble parameter which is too high to allow quark confinement and the generation of an axion potential, one may consider a potential for the axion stemming from another source. Such a possibility involving a dark gauge sector which also breaks the $U(1)_{\text{PQ}}$ has been proposed [53, 54]. The bias mechanism would still result in the accumulation of the axion field at the minima of this potential which would carry over to the time of QCD confinement because the axion field is weakly coupled. This would solve the domain wall problem unless the two potentials are correlated and overlap. If such a potential is generated early on in the deSitter universe, the axion field is correlated on super-horizon scales after reheating and the subsequent PQ breaking does not result in the generation of domain walls within our particle horizon. In this sense the scenario mimics the

---

[2]However, if the walls are connected by strings, the biased initial conditions mean that the initial string configuration is energetically disfavoured and would likely relax to an unbiased state by emitting axions *before* the formation of DWs. Moreover recent lattice simulations show that taking superhorizon inflationary correlations into account can undermine the bias mechanism [39]. In any case we find here that bias does not solve the axion DW problem.

pre-inflationary PQ breaking scenario. Should the potential be generated at the end of inflation with a Hubble parameter large enough to result in a steady state distribution, then the subsequent domain walls are biased and decay, unless more than one minimum of the two potentials overlaps. Note however, that the two scales $f_{PQ}$ and $H_{infl}$ must be closely aligned to achieve this goal.

## 3  Domain wall decay

When the string-wall network decays, its energy density is released as gravitational radiation and axions. However there are cosmological constraints on dumping a large amount of energy via either of these decay channels.

We define the density parameter in any component as the ratio of its energy density to the critical energy density of the universe expanding at a rate $H_0 \sim 70 \text{ km s}^{-1}\text{Mpc}^{-1}$ today [1]:

$$\Omega_X(t) \equiv \frac{\rho_X(t)}{\rho_{\text{crit}}}, \quad \rho_{\text{crit}} = \frac{3H_0^2 M_{\text{Pl}}^2}{8\pi} \simeq 3.8 \times 10^{-47} \text{ GeV}^4. \tag{18}$$

Strings appear at the PQ breaking scale $T \simeq v_a$ with energy density (in the scaling regime),

$$\rho_{\text{str}}(t) = \mathcal{A}_{\text{str}}(t)\frac{\mu_{\text{str}}}{t^2} = \mathcal{A}_{\text{str}}(t)\frac{\pi v_a^2 \ln(v_a t)}{t^2}, \tag{19}$$

where $\mathcal{A}_{\text{str}}(t)$ is the number of strings per Hubble patch and $\mu_{\text{str}}$ is the string tension. Efficient cutting of the strings suggests $\mathcal{A}_{\text{str}}(t) \sim 1$ [55]. Domain walls form when the axion mass becomes dynamically important, overcoming the Hubble drag, at $t_{\text{DW}} \sim m_a^{-1}$. They connect each string to $N_{\text{DW}}$ walls which have energy density (in the scaling regime):

$$\rho_{\text{DW}}(t) = \mathcal{A}_{\text{DW}}(t)\frac{\sigma_{\text{DW}}}{t} \simeq \mathcal{A}_{\text{DW}}(t)\frac{9m_a f_{PQ}^2}{t}. \tag{20}$$

The ratio of the energy density in domain walls and strings is thus

$$\frac{\Omega_{\text{DW}}(t_{\text{DW}})}{\Omega_{\text{str}}(t_{\text{DW}})} = \frac{\rho_{\text{DW}}(t_{\text{DW}})}{\rho_{\text{str}}(t_{\text{DW}})} \simeq \frac{9}{N_{\text{DW}}\pi \ln(v_a t)} \simeq 4.7 \times 10^{-2} N_{\text{DW}}^{-1}. \tag{21}$$

Soon after the formation of domain walls, the string-wall network is dominated by the dynamics of the walls which freely drag the strings around after a time

$$t_{\text{DWdom}} = \frac{\mathcal{A}_{\text{str}}(t)}{\mathcal{A}_{\text{DW}}(t)}\frac{\mu_{\text{str}}}{\sigma_{\text{DW}}} \simeq \frac{\pi}{9}N_{\text{DW}}t_{\text{DW}}\ln(v_a t) \simeq 21 N_{\text{DW}}t_{\text{DW}}\left(\frac{\log(v_a t_{\text{DWdom}})}{60}\right). \tag{22}$$

Once the walls dominate, the string contribution to the energy density is negligible and may be ignored.

### 3.1  Decay into relativistic particles

Since the wall network decays exponentially fast, treating the decay as instantaneous is a justified simplification. If the decay is mainly into gravitational waves or if the axions are light enough to remain relativistic till today, the energy density of domain walls is converted into, and remains, radiation. The usual radiation energy density at $T \ll m_e$ is

$$\rho_{\text{rad}} = \rho_\gamma\left(1 + \frac{7}{8}\left(\frac{4}{11}\right)^{4/3} N_{\text{eff}}\right), \tag{23}$$

where $\rho_\gamma = \pi^2 T^4/15$ is the energy density in photons and $N_{\text{eff}} = 3.046$ is the effective number of neutrino species. Any additional relativistic energy density $\rho$ contributes an equivalent number of effective neutrino species:

$$\Delta N_{\text{eff}} \equiv N_{\text{eff}} - 3.046 = \frac{8}{7}\left(\frac{11}{4}\right)^{4/3}\frac{\rho}{\rho_\gamma}. \tag{24}$$

This is bounded by the Planck limit on 'dark radiation' from observations of CMB anisotropies: $\Delta N_{\text{eff}} \lesssim 0.3$ [56], hence we have $\rho/\rho_\gamma < 0.07$. Because the ratio of the energy density of domain walls and radiation scales $\propto R(t)^2$, this requires the walls to decay long before they come to dominate.

With the usual time-temperature relationship, $t(T) = 0.77\,\text{s}\,(g_*(T)/10)^{-1/2}\,(T/\text{MeV})^{-2}$ in terms of $g_*(T)$ the effective relativistic degrees of freedom [48], we find

$$\frac{\Omega_{\text{DW}}(t)}{\Omega_{\text{rad}}(t)} = 0.01\left(\frac{t}{\text{s}}\right)^{-1}\left(\frac{g_*(T(t))}{10}\right)^{-1}\left(\frac{T(t)}{\text{MeV}}\right)^{-4}\left(\frac{f_{\text{PQ}}}{10^{12}\,\text{GeV}}\right) \lesssim 0.04, \tag{25}$$

implying that the domain walls decaying into radiation must do so before

$$t_{\text{dec}} \lesssim 2.4\,\text{s}\left(\frac{f_{\text{PQ}}}{10^{12}\,\text{GeV}}\right)^{-1}. \tag{26}$$

This reproduces our constraint (16) obtained by assuming a decay time around $t_{\text{BBN}} \sim 1\,\text{s}$.

## 3.2 Decay into particles that become non-relativistic

Again we will assume the wall decay to be instantaneous but all the energy density to be converted into axions which subsequently become non-relativistic. Now

$$\Omega_a^{\text{DW}}(t) = \frac{\omega_a}{\rho_{\text{crit}}}n_a^{\text{DW}} = \frac{\omega_a}{\langle\omega_a\rangle}\Omega_{\text{DW}}(t_{\text{dec}})\left(\frac{R(t_{\text{dec}})}{R(t)}\right)^3, \tag{27}$$

with

$$\frac{\omega_a}{\langle\omega_a\rangle} = \frac{1 + (K-1)R(t_a)/R(t)}{K} = \frac{1 + (K-1)\sqrt{t_a/t}}{K}, \tag{28}$$

where $\omega_a$ is the axion energy and $\langle\omega_a\rangle$ its average for the radiated axions, $K$ is the axion kinetic energy (in units of $m_a$) and $t_a$ the time at which the axion in question was radiated. Numerical studies find $K \sim 100$ [17][3] i.e. the axions are initially highly relativistic and scale like radiation, but after the universe expands sufficiently they turn non-relativistic and behave like matter which is decoupled from the thermal bath. To ensure that their present energy density does not exceed that of dark matter, the wall network must decay early enough. We know that the universe is radiation dominated during BBN and we assume this to be so during the entire period when domain walls are present. The axions from wall decay become non-relativistic at

$$t_{\text{nr}} \gtrsim t_{\text{dec}}(K-1)^2. \tag{29}$$

Subsequently their energy density scales as matter which is limited by the dark matter abundance today [1]:

$$\Omega_a^{\text{DW}}(t_0) \simeq \frac{1}{K}\Omega_{\text{DW}}(t_{\text{dec}})\left(\frac{R(t_{\text{dec}})}{R(t_{\text{eq}})}\right)^3\left(\frac{R(t_{\text{eq}})}{R(t_0)}\right)^3 \leq x_a\Omega_{\text{DM}} \simeq 0.26x_a, \tag{30}$$

---

[3]Recent studies [57, 58] find $K$ to be of $\mathcal{O}(1)$ in which case the decay axions turn non-relativistic earlier, leading to a stronger constraint (see Figure 3), so we are being conservative here.

with $t_{\text{eq}} \simeq 3.3 \times 10^{36} \, \text{GeV}^{-1}$ the time of matter-radiation equality and $x_a$ the fraction of dark matter in the *non*-thermal axions from wall decay. The most conservative estimate is $x_a = 1$, i.e. all the dark matter is contributed by axions from wall decay. Considerations of structure formation probably impose a much stronger limit.[4] This requires

$$t_{\text{dec}} < 1.0 \times 10^{18} \, \text{GeV}^{-1} \left( \frac{K}{100} \right)^2 \left( \frac{f_{\text{PQ}}}{10^{12} \, \text{GeV}} \right)^{-2} \simeq 0.66 \, \mu\text{s}, \tag{31}$$

which is significantly earlier than the constraint of $t \lesssim 50\,\text{s}$ from wall domination. The corresponding temperature is $\sim 2\Lambda_{\text{QCD}}$ when the walls have just about formed, so this bound cannot in fact be improved any further.

Hence the range (15) of the symmetry-breaking scale for which a tilt in the potential can solve the domain wall problem without spoiling the PQ solution to the strong-*CP* problem reduces to

$$f_{\text{PQ}} \lesssim 2.2 \times 10^9 \, \text{GeV}. \tag{32}$$

This rules out the tilt solution to the domain wall problem for a QCD axion which is lighter than a few meV.

Should the decay be induced by a tilt from a higher dimensional operator (§ 2.1), then the decay time is given by $t_{\text{dec}} \simeq t_{\text{DW}} \mu^{-1}$. The above constraint then tightens to

$$f_{\text{PQ}} \lesssim 1.8 \times 10^8 \, \text{GeV}. \tag{33}$$

Since the dimensionality of the operator must be integer, the constraint is even tighter:

$$f_{\text{PQ}} \lesssim 7.6 \times 10^7 \, \text{GeV}. \tag{34}$$

To be more precise we now drop the assumption of instantaneous decay and consider decay into both axions and gravitational waves in order to obtain a robust constraint.

### 3.3 Generalised decay

The coupled equations governing the decay of the string-wall network are [58, 59]:

$$\frac{\partial \Omega_{\text{DW}}}{\partial t} = -H(t)\Omega_{\text{DW}} - \frac{\partial \Omega_{\text{DW} \to a}(t)}{\partial t} - \frac{\partial \Omega_{\text{DW} \to \text{GW}}}{\partial t}, \tag{35}$$

$$\frac{\partial n_a^{\text{DW}}(t)}{\partial t} = -3H(t)n_a^{\text{DW}}(t) + \frac{\rho_{\text{crit}}}{\langle \omega_a \rangle} \frac{\partial \Omega_{\text{DW} \to a}(t)}{\partial t}, \tag{36}$$

$$\frac{\partial \Omega_{\text{GW}}}{\partial t} = -4H(t)\Omega_{\text{GW}} + \frac{\partial \Omega_{\text{DW} \to \text{GW}}}{\partial t}, \tag{37}$$

with $\Omega_{\text{DW} \to a}$ and $\Omega_{\text{DW} \to \text{GW}}$ the instantaneous energy density converted from walls to axions or gravitational waves, $n_a^{\text{DW}}(t)$ the number density of radiated axions and $\langle \omega_a \rangle$ their average energy, and $\Omega_{\text{GW}}$ the density parameter of gravitational waves.

We begin by estimating the gravitational wave radiation from domain walls oscillating at a typical frequency dictated by their size. By the quadruple formula, the power in gravitational waves radiated by a domain wall of size $\ell$ oscillating at the typical frequency $\ell^{-1}$ is [16, 17]

$$P_{\text{GW}} \simeq \frac{\sigma_{\text{DW}}^2}{M_{\text{Pl}}^2} \ell^2. \tag{38}$$

---

[4]If $K$ is small the decay axions have a short free streaming length and may thermalise [27]; there are then no constraints from observations of the Lyman-$\alpha$ forest, however this requires further investigation.

A wall bubble has energy $\rho \sim \sigma_{\text{DW}}\ell^2 H^3$ and if the number of bubbles stays constant then the typical size of the bubbles is $\ell^2 \sim \rho_{\text{DW}} N_{\text{DW}}/\sigma_{\text{DW}} H^3$, which then suggests

$$\frac{\partial \Omega_{\text{DW}\to\text{GW}}}{\partial t} \simeq \frac{H^3}{\rho_{\text{crit}}} P_{\text{GW}} \simeq \frac{\sigma_{\text{DW}}}{M_{\text{Pl}}^2} N_{\text{DW}} \Omega_{\text{DW}}. \tag{39}$$

The scaling of the DW energy density (3), which in physical coordinates reads:

$$\rho_{\text{DW}} \propto \frac{\sigma_{\text{DW}}}{\eta R(t)} \exp\left[-\mu^3\left(\frac{\eta R(t)}{\eta_{\text{DW}} R(t_{\text{DW}})}\right)^3\right] = \frac{\sigma_{\text{DW}}}{t} \exp\left[-\mu^3\left(\frac{t}{t_{\text{DW}}}\right)^3\right]. \tag{40}$$

Note that accounting for collapsing DW bubbles does not change this scaling significantly [32, 57–59]. Now we can solve the differential equation (37) to find:

$$\Omega_{\text{GW}} \simeq \frac{\sigma_{\text{DW}}^2 N_{\text{DW}}}{3 M_{\text{Pl}}^2 \rho_{\text{crit}}}\left(\frac{t_{\text{DW}}}{t}\right)^2 \left[\text{E}_{1/3}(\mu^3) - \left(\frac{t}{t_{\text{DW}}}\right)^2 \text{E}_{1/3}\left(\mu^3\left(\frac{t}{t_{\text{DW}}}\right)^3\right)\right], \tag{41}$$

where $\text{E}_n(x) = \int_1^\infty e^{-xt}/t^n \mathrm{d}t$ is the exponential integral function. A detailed numerical simulation [57] finds a value that is higher by a factor of 5 or so.

We can also solve eq.(36) by substitution of eq.(35) to find:

$$\begin{aligned}
\Omega_a^{\text{DW}} \simeq {} & \frac{\sigma_{\text{DW}}}{t_{\text{DW}}\rho_{\text{crit}}} \frac{\omega_a}{\langle\omega_a\rangle}\left(\frac{t_{\text{DW}}}{t}\right)^{\frac{3}{2}}\left[e^{-\mu^3} - \sqrt{\frac{t}{t_{\text{DW}}}}e^{-\mu^3\left(\frac{t}{t_{\text{DW}}}\right)^3}\right] \\
& + \frac{\sigma_{\text{DW}}}{t_{\text{DW}}\rho_{\text{crit}}} \frac{\omega_a}{\langle\omega_a\rangle}\left(\frac{t_{\text{DW}}}{t}\right)^{\frac{3}{2}}\left[\frac{1}{3}\text{E}_{5/6}(\mu^3) - \frac{1}{3}\sqrt{\frac{t}{t_{\text{DW}}}}\text{E}_{5/6}\left(\mu^3\left(\frac{t}{t_{\text{DW}}}\right)^3\right)\right] \\
& + \frac{\sqrt{\pi}\sigma_{\text{DW}}^2 N_{\text{DW}}}{3 M_{\text{Pl}}^2 \mu^{3/2}\rho_{\text{crit}}} \frac{\omega_a}{\langle\omega_a\rangle}\left(\frac{t_{\text{DW}}}{t}\right)^{\frac{3}{2}}\left[\text{erf}(\mu^{3/2}) - \text{erf}\left(\mu^{3/2}\left(\frac{t}{t_{\text{DW}}}\right)^{3/2}\right)\right]. \tag{42}
\end{aligned}$$

In the following we will assume that most axions are produced at wall decay, so we can replace $t_a$ with $t_{\text{dec}} \sim t_{\text{DW}}/\mu$ in eq. (28). This is a good approximation given the exponentially fast decay of the domain wall network.

The gravitational radiation (41) produced by the decay of the DW is subject to the same $N_{\text{eff}}$ bound as before:

$$\lim_{t\to\gg t_{\text{dec}}} \frac{\Omega_{\text{GW}}}{\Omega_\gamma} \simeq \frac{5}{\pi} \frac{\sigma_{\text{DW}}^2 N_{\text{DW}}}{M_{\text{Pl}}^2 \text{MeV}^4}\left(\frac{t_{\text{DW}}}{0.77\,\text{s}}\right)^2 \left(\frac{10}{g_*}\right)\text{E}_{1/3}\left(\mu^3\right) \lesssim 0.07. \tag{43}$$

We conclude that this is true iff

$$f_{\text{PQ}} \lesssim 7\times 10^{10}\,\text{GeV}\left(\frac{\text{E}_{1/3}\left(\langle\theta\rangle^3\right)}{\text{E}_{1/3}\left(\mu^3\right)}\right)^{3/8}, \tag{44}$$

where we have used eq.(9) to estimate

$$\mu = |g|\left(\frac{M_{\text{Pl}}}{m_a}\right)^2 \left(\frac{f_{\text{PQ}}}{\sqrt{2}M_{\text{Pl}}}\right)^{2m+n-2} \simeq \frac{\langle\theta\rangle}{\sin\delta}\frac{N_{\text{DW}}}{n} < 10^{-10}\left(\frac{\langle\theta\rangle}{10^{-10}}\right)\frac{N_{\text{DW}}}{n}\frac{1}{\sin\delta}. \tag{45}$$

There is some theoretical uncertainty in interpreting the experimental bound on the neutron EDM; we have used a more conservative constraint on $\langle\theta\rangle$ than in refs. [42, 57, 58].

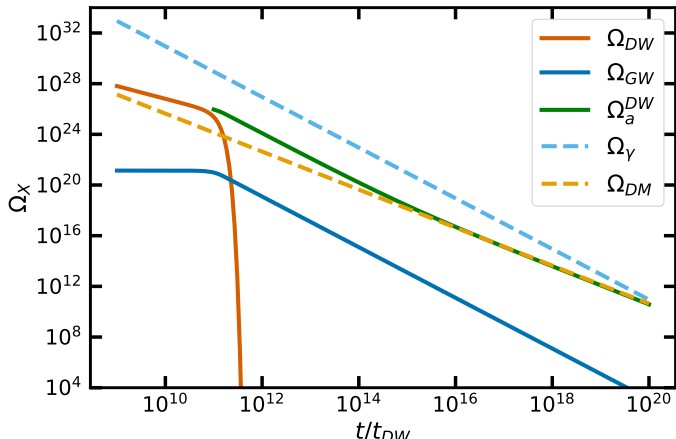

Figure 2: Evolution of various components of the energy density with time (in units of $t_{\rm DW} = m_a^{-1}$), for a tilt parameter $\mu = 10^{-11}$ and $f_{\rm PQ} = 5 \times 10^7$ GeV. The dashed blue (orange) line indicates the usual radiation (dark matter) content. The orange line indicates the axion domain walls which decay at $t_{\rm dec} \sim t_{\rm DW}/\mu$, and the blue and green lines correspond to their decay products, respectively gravitational waves and axions (taking $K = 100$). The latter turn non-relativistic at $t_{\rm nr} \sim t_{\rm dec}K^2$ and are conservatively assumed to contribute the present dark matter abundance.

Assuming instantaneous decay of the domain walls, the peak frequency of gravitational waves can be estimated from the Hubble scale at the decay epoch [60]

$$
f_{\rm GW} \simeq H(t_{\rm dec})\left(\frac{R(t_{\rm dec})}{R(t_{\rm eq})}\right)\left(\frac{R(t_{\rm eq})}{R(t_0)}\right)
$$

$$
\simeq 1.4 \times 10^{-11}\,{\rm Hz}\left(\frac{g_*(T_{\rm dec})}{10}\right)^{1/2}\left(\frac{g_{*s}(T_{\rm dec})}{10}\right)^{-1/3}\left(\frac{T_{\rm dec}}{\rm MeV}\right), \tag{46}
$$

while the amplitude is from eq.(41),

$$
\lim_{t \to t_0} \Omega_{\rm GW} \simeq 1.2 \times 10^{-9}\left(\frac{f_{\rm PQ}}{10^{12}\,{\rm GeV}}\right)^{8/3}\left(\frac{E_{1/3}\left(\mu^3\right)}{E_{1/3}(\langle\theta\rangle^3)}\right). \tag{47}
$$

A far more substantial contribution is made however by the radiated axions which contribute to the dark matter abundance today. Requiring that the total axion abundance not exceed the latter, we obtain the severe constraint:

$$
\lim_{t \to t_0} \Omega_a^{\rm DW} < 0.26 x_a \quad \Rightarrow \quad f_{\rm PQ} \lesssim 3.3 \times 10^8 x_a^{6/7}\,{\rm GeV} \text{ i.e. } m_a \gtrsim 17\, x_a^{-6/7}\,{\rm meV}. \tag{48}
$$

Our bound is consistent with previous numerical work [42, 58]. Figure 3 shows how this constraint scales with the tilt parameter $\mu \approx \langle\theta\rangle$ (see eq.45). As the experimental limit on the neutron EDM improves, the smaller the tilt allowed, so the above constraint on the QCD axion will tighten even further with forthcoming measurements. We also show the scaling of the bound with $N_{\rm DW}$ in Figure 4. For the DFSZ model in particular $N_{\rm DW} = 6$ so the bound is $m_a \gtrsim 11\, x_a^{-6/7}$ meV i.e. $f_{\rm PQ} \lesssim 5.4 \times 10^8 x_a^{6/7}$ GeV.

The above constraint used the conservative value $x_a = 1$, i.e. the dark matter is taken to be entirely constituted of non-relativistic axions which were born relativistic but out of thermal equilibrium. Considerations of structure formation place constraints on axion 'hot dark matter' assuming the axions initially have a Bose-Einstein distribution [61, 62]. Solution

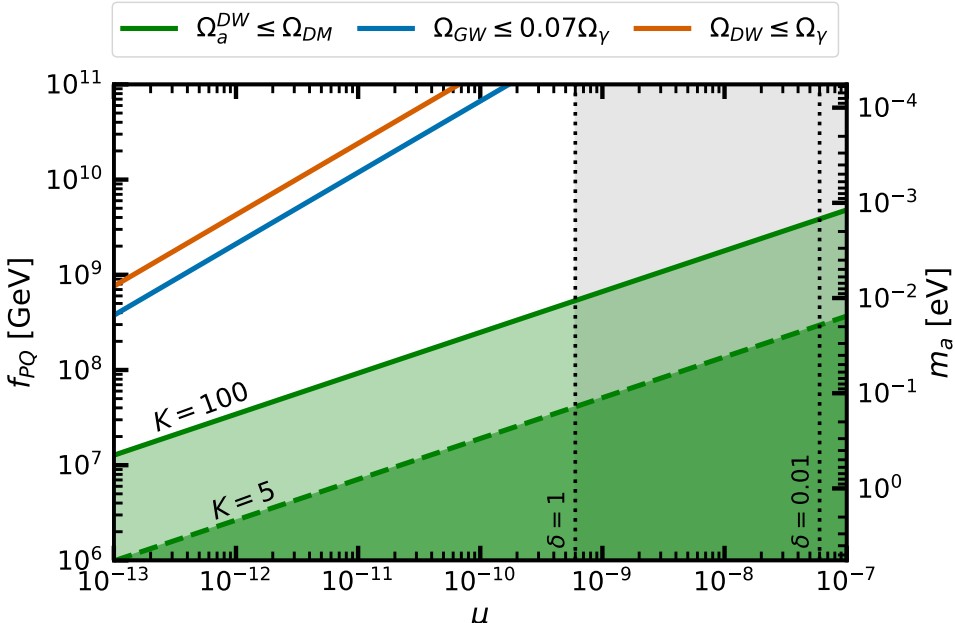

Figure 3: Scaling of the upper bound on $f_{PQ}$ (and corresponding lower bound on $m_a$) with the tilt parameter $\mu$ (eq.45) assuming QCD axions from the decay of domain walls make up all the dark matter; the region *above* the green curve is excluded. The experimental limit on the neutron EDM requires $\mu < 10^{-10} N_{DW}$ (vertical dotted line, taking $N_{DW} = 6$), i.e. $m_a \gtrsim 11$ meV. If $\delta$ is fine-tuned to be $10^{-2}$, then $\mu$ increases to $10^{-8} N_{DW}$ thus allowing the grey shaded region, i.e. lighter axions down to $m_a \sim 1.5$ meV. However if the decay axions are only mildly relativistic (see eq. 28) with $K \sim 5$ [58] rather than $K = 100$ as assumed above, this yields a stronger constraint (dashed green curve) which implies $m_a \gtrsim 139$ meV for $\delta = 1$, so the previous bound is quite robust.

of the relevant Boltzmann equations governing decoupling yields the actual distribution; this imposes a restrictive upper bound $m_a < 0.24$ eV [63]. The relevant QCD (DFSZ) axion window is then $\sim 11 - 240$ meV although this will narrow further if $x_a$ can be constrained to be below unity from considerations of structure formation. Whereas axions of such mass are subject to constraints on stellar energy loss [64], the most stringent such bound from SN 1987a is significantly weakened taking astrophysical uncertainties into account [65]. In fact there are indications of anomalous stellar cooling which would indicate a mass of $\mathcal{O}(10)$ meV for the QCD axion [66].

## 4 Conclusion

We have revisited the cosmological domain wall problem which poses a serious threat to the QCD axion. If the Peccei-Quinn symmetry breaks after inflation, then in any model with $N_{DW} > 1$ quarks charged under $U(1)_{PQ}$, a network of domain walls is created and comes to dominate over both radiation and matter. Such an universe undergoes power-law inflation without end, incompatible with the universe we observe today.

Finding a mechanism which can make the walls collapse is challenging. A statistical bias induced by inflation in the population of the vacuum states results in exponential decay of the wall network, however, the vast separation of the relevant scales in the field space — $f_{PQ}, H_{infl}$,

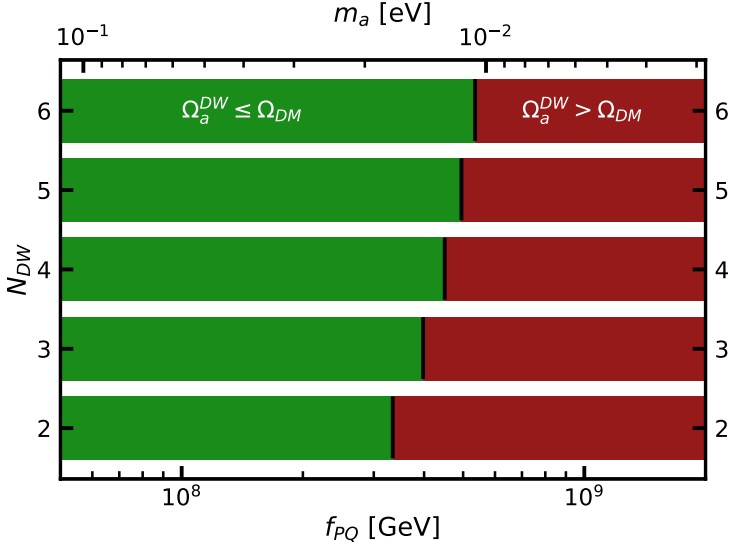

Figure 4: Scaling of the upper bound on $f_{\text{PQ}}$ (and corresponding lower bound on $m_a$) with $N_{\text{DW}}$ for the QCD axion in the post-inflationary scenario.

and $\Lambda_{\text{QCD}}$ — means that the necessary bias cannot be generated.

The other known way to make the wall network decay is to introduce a small tilt in the potential. However thus explicitly breaking the symmetry leads to the reappearance of the strong-$CP$ problem, hence the tilt is limited by the experimental upper limit on the neutron EDM. If the tilt results, as is usually assumed, from a non-renormalisable, Planck scale suppressed operator reflecting violation of the global $U(1)_{\text{PQ}}$ symmetry by quantum gravity effects, then its dimension is tightly constrained. As measurements of the neutron EDM improve further, this window will eventually close altogether.

Independent of the mechanism, the energy density in the domain walls is released as gravitational waves and relativistic axions which subsequently turn non-relativistic, both of which are constrained by observations. We find that QCD axion models with $N_{\text{DW}} > 1$ and post-inflationary Peccei-Quinn breaking are severely constrained, e.g. for the DFSZ axion with $N_{\text{DW}} = 6$, we require $f_{\text{PQ}} \lesssim 5.4 \times 10^8$ GeV, i.e. $m_a \gtrsim 11$ meV. Experimental searches must thus focus on higher mass axions which are cosmologically still viable, and can have noticeable effects on stellar energy loss.

# Acknowledgements

We thank Prateek Agrawal and Tomasz Krajewski for discussions and Pierre Sikivie for encouragement. We are grateful to Andreas Ringwald for several helpful suggestions and drawing our attention to previous related work. We also thank the Referees and handling editor for constructive remarks and criticism.

**Funding information** SS is a member of the 'Quantum Sensors for the Hidden Sector' consortium funded by the UK Science & Technology Facilities Council.

# A    Axion field evolution

## A.1    Inflationary universe

During cosmic inflation the universe undergoes rapid expansion characterised by an approximately constant Hubble parameter $H_{\text{infl}}$. This implies an exponential growth of the scale factor $R(t) \propto \exp(H_{\text{infl}}t)$ and the metric is approximately of the deSitter form

$$ds^2 = R(t)\eta_{i,j}dx^i dx^j, \tag{A.1}$$

with $\eta$ the flat Minkowski metric

The dynamics of the axion field $\phi(x) \in [0, 2\pi f_{\text{PQ}})$ is governed by the semi-classical equation of motion (EOM)

$$\left[\left(\frac{\partial}{\partial t}\right)^2 + 3H_{\text{infl}}\frac{\partial}{\partial t} - \frac{1}{a(t)^2}\nabla^2\right]\phi(x) + V'(\phi(x)) = 0, \tag{A.2}$$

with potential

$$V(\phi(x)) = m_a^2 f_{\text{PQ}}^2\left(1 - \cos\left(N\frac{\phi(x)}{f_{\text{PQ}}}\right)\right). \tag{A.3}$$

The mass term in the potential is time-dependent and has been calculated on the lattice [13] — see Eq.(1).

During deSitter expansion, fields have two naturally separated scales, sub- and super-horizon, with physical momentum $k > H_{\text{infl}}^{-1}$ and $k < H_{\text{infl}}^{-1}$ respectively. While the field is frozen on super-horizon scales, on sub-horizon scales it evolves according to eq. (A.2). Sub-horizon scales are said to 'exit the horizon' when their physical momenta $k = p/R(t)$ are sufficiently redshifted by the expansion. Following Refs. [49,50] we write the axion field as a mode expansion for sub-horizon modes, and for super-horizon modes a coarse-grained fluctuation field $\chi$, averaged over many horizon sizes $\bar{\chi}$:

$$\phi(x) = \int \Theta\left(p - \varepsilon H_{\text{infl}}e^{H_{\text{infl}}t}\right)\left[\hat{a}_{\mathbf{p}}\phi_{\mathbf{p}}(t)e^{i\mathbf{p}\cdot\mathbf{x}} + \text{h.c.}\right] + \chi(x) - \bar{\chi}. \tag{A.4}$$

The fluctuation field obeys a Langevin-type equation

$$\frac{\partial\phi}{\partial t} = \frac{1}{3H_{\text{infl}}}\left(\frac{\nabla^2\phi}{e^{2H_{\text{infl}}t}} - \frac{\partial V}{\partial\phi}\right) + \eta(x), \tag{A.5}$$

with $\eta(\mathbf{x}, t)$ acting as white noise sourced by the modes leaving the horizon, which causes the averaged field to random walk. The Langevin equation can be translated into a Fokker-Planck equation for the normalised probability distribution of the field $P(\chi, \bar{\chi}, t)$:

$$\frac{\partial P(\chi, \bar{\chi}, t)}{\partial t} = \frac{\partial}{\partial\chi}\left(\frac{1}{3H_{\text{infl}}}\frac{\partial V}{\partial\chi}P(\chi, \bar{\chi}, t)\right) + \frac{H_{\text{infl}}^3}{8\pi^2}\frac{\partial^2 P(\chi, \bar{\chi}, t)}{\partial\chi^2}. \tag{A.6}$$

The solution is given by [67]

$$P(\chi, \bar{\chi}, t) = \exp\left(-\frac{4\pi^2}{3H_{\text{infl}}^4}V(\chi)\right)\sum_{n=0}^{\infty}a_n\Phi_n(\chi)\exp\left(-\Lambda_n(t - t_i)\right), \tag{A.7}$$

with $\Phi_n(\chi)$ the eigenfunctions of

$$-\frac{1}{2}\frac{\partial^2\Phi_n}{\partial\chi^2} + \frac{1}{2}\left(\left(\frac{4\pi^2}{3H_{\text{infl}}^4}\frac{\partial V}{\partial\chi}\right)^2 - \frac{4\pi^2}{3H_{\text{infl}}^4}\frac{\partial^2 V}{\partial\chi^2}\right)\Phi_n = \frac{4\pi^2\Lambda_n}{H_{\text{infl}}^3}\Phi_n, \tag{A.8}$$

while the coefficients $a_n$ are given by the initial condition at $t = t_i$ as

$$a_n = \int P(\chi, \bar{\chi}, t) \Phi_n(\chi) \exp\left( \frac{4\pi^2}{3H_{\text{infl}}^4} \frac{\partial V}{\partial \chi} \right) d\chi. \tag{A.9}$$

The distribution (A.7) evolves towards the stationary solution for late times and thus for the average field

$$P_{\text{stationary}}(\bar{\chi}) = \exp\left( -\frac{4\pi^2}{3H_{\text{infl}}^4} V(\bar{\chi}) \right) \Big/ \int \exp\left( -\frac{4\pi^2}{3H_{\text{infl}}^4} V(\chi) \right) d\chi. \tag{A.10}$$

For the axion potential (A.3), Eq.(A.8) reduces to a Schrödinger type equation after neglecting terms of $\mathcal{O}(m_a/H_{\text{infl}})^4$:

$$\frac{\partial^2 \Phi_n}{\partial \chi^2} + \left[ \frac{4\pi^2}{3} \frac{N^2 m_a^2}{H_{\text{infl}}^4} \cos\left( N \frac{\chi}{f_{\text{PQ}}} \right) + \frac{8\pi^2 \Lambda_n}{H_{\text{infl}}^3} \right] \Phi_n. \tag{A.11}$$

The solution is a Mathieu function:

$$M_C\left( \frac{32\pi^2 f_{\text{PQ}}^2}{N^2 H_{\text{infl}}^3} \Lambda_n, \frac{8\pi^2 f_{\text{PQ}}^2 m_a^2}{3H_{\text{infl}}^4}, N \frac{\chi}{2f_{\text{PQ}}} \right) \approx \cos\left( \sqrt{\frac{8\pi^2 f_{\text{PQ}}^2}{H_{\text{infl}}^3} \Lambda_n} \frac{\chi}{f_{\text{PQ}}} \right), \tag{A.12}$$

where we have used $m_a \ll f_{\text{PQ}}, H_{\text{infl}}$ in the last step. The eigenvalues are then given by

$$\Lambda_n \approx \frac{n^2}{8\pi^2} \left( \frac{H_{\text{infl}}}{f_{\text{PQ}}} \right)^2 H_{\text{infl}}, \tag{A.13}$$

and the first correction to the stationary solution (A.10) is exponentially suppressed when the number of e-folds exceeds $8\pi^2(f_{\text{PQ}}/H_{\text{infl}})^2$. Thus, the smaller the inflationary scale, the longer it takes to reach the stationary state.

Assuming the potential admits slow-roll, the solution for the fluctuation field can be written as a Gaussian wrapped around the compact field region:

$$\begin{aligned} P(\chi, \bar{\chi}, t) &= \sum_{k=-\infty}^{\infty} \sqrt{\frac{2\pi}{H_{\text{infl}}^3 t}} \exp\left( -\frac{2\pi^2}{H_{\text{infl}}^3 t} \left( \chi - \bar{\chi} + 2\pi f_{\text{PQ}} k \right)^2 \right) \\ &= \frac{1}{2\pi} \vartheta\left( \frac{\theta - \bar{\theta}}{2}, \exp\left[ -\frac{H_{\text{infl}}^3 t}{16\pi^3 f_{\text{PQ}}^2} \right] \right), \end{aligned} \tag{A.14}$$

with $\vartheta$ the Jacobi theta function and $\theta = \chi/f_{\text{PQ}}$. Physically, two effects are competing here. On the one hand, each mode leaving the horizon gives the distribution a kick of $\mathcal{O}(H_{\text{infl}}/2\pi)$, thus widening the distribution. On the other hand, the potential causes the average to concentrate around the vacuum states $V(\chi) = 0$. This concentration near the potential minima will lead to the appearance of a statistical bias in the population of the states, hence it is clear that our scenario only works when the axion develops a potential *during* inflation.

## A.2 FLRW universe and bias generation

After cosmic inflation and reheating the universe enters a period of radiation-dominated Friedman-Lemaître-Robertson-Walker (FLRW) expansion with a metric similar to eq.(A.1) but with $H(t) = 1/2t$ no longer constant and $R(t) \propto \sqrt{t}$. Accelerated expansion has stopped

so the causal horizon starts growing which leads to the re-entry of scales which had left the horizon in the deSitter phase. The equation of motion changes to

$$\left[\left(\frac{\partial}{\partial t}\right)^2 + 3H(t)\frac{\partial}{\partial t}\right]\phi(x) + V'(\phi(x)) = 0, \qquad (A.15)$$

where we have neglected the spatial derivatives (smoothed out during the inflationary epoch). The initial conditions are given by the inflationary Hubble parameter $1/2t_i = H_{\text{infl}}$ and the field at the end of inflation $\phi_i = \chi(t_i)$; note that $\partial_t \phi_i = 0$ because of inflation. The Hubble drag term decreases until the potential dominates the field evolution. At this point $\phi(x)$ settles into one of the potential minima.

The bias $\varepsilon$ is defined as the difference in the probability of populating the degenerate minima. Qualitatively the bias arises because the averaged distribution during inflation (eq.A.10) is concentrated around the potential minima and the fluctuation field distribution (eq.A.14) has a finite width, making it less probable to populate vacuum states further away from the average field value. For the simplest case $N_{\text{DW}} = 2$ we follow the definition [38]:

$$b(\bar{\chi}) = \int f(\phi_i)P(\phi_i, \bar{\chi}, t_i)d\phi_i, \qquad (A.16)$$

with $f(\phi)$ a function taking the values $-1, 1$ when the evolution of $\phi$ ends in one or the other vacuum state respectively. Inflation gives access only to the probability distribution of $\bar{\chi}$, hence this translates into a probability of finding a bias [38]:

$$P(|b| < x) = \sum_{\bar{\chi}; b = b(\bar{\chi})} \int_{b(\bar{\chi})=-x}^{b(\bar{\chi})=x} P_{\text{stationary}}(\bar{\chi})d\bar{\chi}. \qquad (A.17)$$

Once the bias is established the domain wall network becomes unstable and its energy density exponentially drops according to eq.(4). The energy density from the collapsing network is radiated as axions and gravitational waves.

The width of the distribution (A.14), $\sigma = \sqrt{H_{\text{infl}}^3 t/4\pi^2 f_{\text{PQ}}^2}$, is dictated by the ratio $H_{\text{infl}}/f_{\text{PQ}}$. For $H_{\text{infl}} \gg f_{\text{PQ}}$ the distribution is flat over the field range $\theta \in [-\pi, \pi]$ but when $H_{\text{infl}} \ll f_{\text{PQ}}$, the wrapped normal distribution can be approximated by a Gaussian.

## A.3 Narrow Gaussian ($H_{\text{infl}} \ll f_{\text{PQ}}$)

When $H_{\text{infl}} \ll f_{\text{PQ}}$, the distribution is well approximated by a narrow Gaussian as long as the central value $\bar{\theta}$ is far from $-\pi, \pi$. Due to the symmetries of the potential (A.3) the bias function (A.16) is symmetric around the origin and anti-symmetric around the points $\theta = \pm\pi/2$. It is thus sufficient to concentrate on a subset of the field range given by $[0, \pi/2]$ and the entire function can be reconstructed as

$$b(\bar{\chi}) = \delta_{k,0} \begin{cases} \int_{\frac{\pi}{2}}^{\pi} P\left(\theta, \bar{\theta}, t\right)d\theta - \int_{-\frac{\pi}{2}}^{\frac{\pi}{2}} P\left(\theta, \bar{\theta}, t\right)d\theta, & \bar{\theta} \in \left[0, \frac{\pi}{2}\right], \\ -b\left(\pi - \bar{\theta}\right), & \bar{\theta} \in \left(\frac{\pi}{2}, \pi\right], \\ b\left(-\bar{\theta}\right), & \bar{\theta} \in [-\pi, 0). \end{cases} \qquad (A.18)$$

Here we have used the fact that for a finite potential like eq.(A.3) which allows free states, the fact that inflation dilutes the derivatives implies that after inflation the field simply settles into the vacuum state it started above. Thus, the function $f(\phi)$ is $-1$ when $\phi \in [-\pi/2, \pi/2]$ and $+1$ otherwise.

Because of the symmetries, the sum in Eq.(A.17) yields 2 so the probability is:

$$P(|b| < \varepsilon) = 4 \int_{\pi/2}^{b(\bar{\theta})=\varepsilon} P_{\text{stationary}}(\bar{\theta}) \mathrm{d}\bar{\theta}, \tag{A.19}$$

where the upper integration limit requires inverting Eq.(A.18).

By definition, $P(|b| < 1) = 1$ hence the probability rises at high values of $b$. Also by definition $P(|b| < 0) = 0$. The behaviour in between interpolates between these two extremes. Increasing $H_{\text{infl}}$ increases the width of the Gaussian distribution (A.14) and yields smaller values of the bias. This is easily understood since the bias comes from counting the positions of the central value $\bar{\theta}$, weighted with the distribution (A.10). The narrower the Gaussian, the more likely it is to fall into just one or the other vacuum.

The width increases with time hence after a sufficient number of e-folds of inflation the distribution spreads to cover multiple vacuum states. However, since the step-size of the random walk is set by $H_{\text{infl}}$ while the field range is set by $f_{\text{PQ}}$, the required number of e-folds is enormous, $\propto (f_{\text{PQ}}/H_{\text{infl}})^2 \gg 1$. The causally connected regions of the universe are inflated to much larger scales than the observable universe today and no domain wall problem arises. The situation is thus equivalent to pre-inflationary PQ breaking.

### A.4 Wide Gaussian ($H_{\text{infl}} \gg f_{\text{PQ}}$)

In the opposite limit, the distribution is well approximated as flat over the field range, with only small perturbations:

$$P(\theta, \bar{\theta}, t) \simeq \frac{1}{2\pi}\left(1 + 2\cos\left(\theta - \bar{\theta}\right)\exp\left[-\frac{\sigma^2}{2}\right]\right). \tag{A.20}$$

With the same definition of the function $f(\phi)$, the bias can be found analytically:

$$b(\bar{\theta}) \simeq -\frac{4}{\pi}\exp\left[-\frac{\sigma^2}{2}\right]\cos\left(\bar{\theta}\right). \tag{A.21}$$

Inverting this equation gives the central frequency as a function of the bias $\bar{\theta}(b)$ and hence the upper integration limit for the bias probability (A.19):

$$P(|b| < \varepsilon) = 4 \int_{\pi/2}^{\cos^{-1}\left(-(\pi/4)\varepsilon\exp[\sigma^2/2]\right)} P_{\text{stationary}}(\bar{\theta})\mathrm{d}\bar{\theta}. \tag{A.22}$$

The general behaviour is similar to the narrow Gaussian case, however the probability for a smaller bias is significantly larger for a flat distribution, as might be expected.

The above argument assumes that the axion field is classical and has a potential such that the mean value can accumulate around the vacuum states. However when $H_{\text{infl}} \ll f_{\text{PQ}}$ this is not the case. Quantum fluctuations in the deSitter background are of $\mathcal{O}(H_i)$ and drive the PQ field back to the origin, restoring $U(1)_{\text{PQ}}$. In this sense, the Peccei-Quinn symmetry is never broken during inflation, there is no axion potential, and therefore, no bias is generated.

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
