# Peer review of "Ruling out light axions: the writing is on the wall"

_SciPost Physics, doi:SciPost Phys. 15, 003 (2023)_

## Round 2 · Referee Report · Anonymous (Referee 1) · 2023-1-22

Report

The authors study an important open problem in axion physics, namely the dark matter abundance in the so-called post-inflationary scenario with domain wall number $N_W>1$. The authors consider an alternative to the usual mechanism to avoid domain wall overclosure, based on biased initial conditions, which unfortunately they find does not work. They also estimate the dark matter and gravitational wave abundance, providing a bound on the axion mass, if the U(1) Peccei--Quinn (PQ) symmetry has an explicit breaking (‘tilt’). The paper is sufficiently well written and the grammar is clear. I list below my main comments and concerns on the paper.

The authors assume the PQ violation takes the form of the operator in eq. (5), with a dimensionless coupling $g$ that parameterizes the breaking of the global U(1) symmetry assumed to be of order 1. However, this is not necessarily the case, as for instance these operators could be suppressed by $M_{\rm string}$. Additionally, although we expect gravity to break all global symmetries, we do not know to what amount these are broken; in particular, the size of the higher dimensional operators could be much smaller than order one in Planck units (this is the case in many instances of non-perturbative breaking via gravitational instantons, black holes and wormholes). This would change the conclusion about the allowed operator dimension and axion decay constant in Section 2.1. The authors should mention this and change the discussion appropriately.

I believe that there may also be another problem with postulating initial conditions after inflation containing a bias on the axion field (i.e., effectively, more points where $a=0$). Thus, even if such initial conditions were obtained, they may not solve the domain wall problem. This is because the boundary of domain walls are axion strings, and biased initial conditions mean that the initial string configuration has the fundamental domain $[-\pi v_a, \pi v_a]$ wrapped in a non-uniform way in space. This configuration is energetically unfavorable, and the string system will relax to an unbiased configuration before domain walls dominate, radiating the surplus of energy into axion waves. The authors should take this into account and change the discussion appropriately. It would be interesting to understand if a theory without strings could have the domain walls destroyed before they dominate (but perhaps this is ruled out by the authors’ work).

The dyanamics of the domain walls and string might also be significantly more complex than the authors discuss and, in particular, it might not be the case that the assumptions leading to eqs. (40) and (41) are valid. The authors should discuss how their assumptions (i.e. the domain-wall oscillating bubbles) used in their estimate in eqs. (40) and (41) compare with the other assumptions discussed in the literature, tested for instance in the numerical simulations of [55,56,58], and how they change the bound on the axion mass. Note that the dependence on the symmetry breaking parameter $\mu$ in eqs. (40) and (41) is hidden, at least in part, in the exponential integral functions, which is hard to understand at a first sight and should be clarified.

I believe that changes addressing these concerns are important before the article could be published in Scipost.

Minor comments and typos: - why is it important that ${\rm min}V = 0$ below eq. (5)? - from the discussion at the end of pg. 7, it is not immediately obvious what could concretely be the additional source of the potential that leads to relevant biased initial conditions - in eq. (21) there is a double round bracket - vacuua-> vacua, throughout the paper - at the end of pg. 2: “a much stronger bound obtains” -> “a much stronger bound is obtained” ? - on pg. 3 “the DFSZ axion..” -> “the DFSZ axion.” - $z$ and $x$ in the definition of $E_n$ below eq. (41) do not match

  • validity: -
  • significance: -
  • originality: -
  • clarity: -
  • formatting: -
  • grammar: -

Author:  Subir Sarkar  on 2023-02-07  [id 3317]

(in reply to Report 1 on 2023-01-22)
Category:
answer to question

We thank Reviewer 1 for a careful reading of our paper and for pointing out several typos (which we have now corrected - thanks!)

Concerning the substansive point made, viz. that the scale at which global symmetries are violated by quantum gravity effects (call it M_QG) need not be M_Pl but can e.g. be M_string - this is indeed true. However this will only make our constraints stronger; since we wish to be conservative, we take M_QG to be M_Pl. We also agree that the size of the higher dimensional operators (parameterised by g) can be much smaller than order one for non-perturbative symmetry breaking. However this does not affect the two opposing constraints on f_PQ which we discuss, viz. from above by requiring that the neutron does not acquire an observable EDM, and from below by requiring the domain wall network to disappear sufficiently fast. It affects only their interpretation, as in our eqs.(10) and (14). We can thus address the Reviewer's concern by simply replacing M_Pl by M_QG in these equations. In Fig.1, we have now explicitly said in the caption that the constraint shown on f_PQ is obtained taking M_QG to be M_Pl; we had already shown the effect of taking |g| to be 10^-2 rather than 1. Since the constraint shown can easily be scaled for any other choice of M_QG, we hope this is satisfactory.

The Reviewer may also be right that 'bias' need not solve the domain wall problem when the walls end on strings, as in the case of axions. Indeed if biased initial conditions result in energetically unfavorable configurations, then the string system will relax to an unbiased configuration before the walls come to dominate. However since we have shown that *in any case* bias is not relevant for the axion domain wall problem (because of the large separation of scales between f_PQ and \Lambda_QCD when the walls form), we do not discuss this here further - it does however merit further study. We have added footnote 2 to this effect in Section 2.2.

The Reviewer wishes to understand better whether our assumption that the wall network decays exponentially is justfied. Indeed the dynamics is quite complex; as seen even in our early simulation [31] (Fig.3) the co-moving domain wall energy density is not a smooth exponential with conformal time (as in eq.3) but exhibits 'bounces’ as bubbles of the disfavored vacuum collapse and radiate away the energy contained in the walls. This affects the total energy density very little however, and subsequent more detailed simulations [55,56,58] confirm this. However our estimate of the radiated gravitational waves uses the quadrupole formula which strictly applies only to a bubble oscillating at its natural frequency set by its size. The more detailed calculation [56] suggests the value can be higher by up to a factor of ~5, but given that energy density of the gravitational waves generated is sub-dominant by a factor of ~10^6, this does not affect our bound on the axion mass. We have made appropriate comments to this effect in Section 3.3. We also agree that the dependence on the symmetry breaking parameter μ in eq.(42), which follows from eqs.(40) and (41), is hidden in the exponential integral functions - this is why we have shown explicitly the dependence of the derived constraint on μ in Fig.3.

As for the Reviewer's question "why is it important that minV=0 below eq. (5)?", this is of course just the notorious Cosmological Constant problem - we are simply highlighting what is usually brushed under the carpet!

Finally the Reviewer notes that "from the discussion at the end of pg. 7, it is not immediately obvious what could concretely be the additional source of the potential that leads to relevant biased initial conditions". We agree - however this issue is beyond the scope of the present work. We have cited interesting ideas by Ferrer et al. [51] and Caputo & Reig [52] in this context. However as we have already remarked the necessary alignment of f_PQ and H_infl is quite challenging.

---

## Round 2 · Referee Report · Anonymous (Referee 2) · 2023-1-23

Report

The authors discussed the axion model with the domain wall (DW) number $N_{DW}>1$ with biased potential/initial conditions. By using the formula (4) of the DW decay rate to simulate the DW network decaying into axion+gravitaitonal waves evolution via the equations from number/energy conservations (35,36,37), they found that the resulting axion is too much to be consistent with the astrophysical bound.

The implicitly assumed philosophy is naturalness. Namely, the solution of the fine-tuning problem for the strong CP problem should not contain another fine-tuning problem in the forms of the Planck suppressed terms to let the DW decay. With this philosophy, with which I agree, a large decay rate of the DW network is in contradiction to the neutron electric-dipole moment experimental bound.

Of course, there can be some extensions to solve the problem (e.g., a relatively simple one may be an entropy dilution for the relativistic axions soon after the DW collapsing, i.e., DW network is formed and destroyed during the reheating),
I consider the paper interesting and important, showing the natural solution of the strong CP problem, with $N_{DW}>1$ and Peccei-Quinn symmetry breaking after the inflation, is strongly disfavored.

One comment is about their choice of $x_a$, the fraction of the DW induced axion abundance to the total dark matter abudance.
I consider it is more conservative to take $x_a=1$, i.e., it is the dominant dark matter. This is because, the axon produced with $K=100$ at, say, $T_{dec}\sim 1MeV$, becomes non-relativistic around $T=10$ keV. Naively, one can estimate that such axions have free-streaming length $\lesssim 0.01$ Mpc which is smaller than the Lyman-alpha constraint,$ \sim O(0.1)$ Mpc, and I consider that the component can be the dominant dark matter. In addition, there may be some thermalization during the redshift of the axion momentum. This is due to the self-quartic coupling, and Bose enhancement see, e.g., the discussions around Eq. 135 of the Axion cosmology by David Marsh, 1510.07633. Then the structure formation bound may be even weaker. Thus, I consider it should be conservative to take $x_a=1$. Since the authors claim the bound is robust, I would suggest the authors use $x_a=1$ for their sample value and conclusions.

Other than this, I am happy to recomend the paper be published.

Requested changes

  1. use $x_a=1$ instead for the conservative bound.

  • validity: high
  • significance: top
  • originality: good
  • clarity: top
  • formatting: excellent
  • grammar: good

Author:  Subir Sarkar  on 2023-02-07  [id 3318]

(in reply to Report 2 on 2023-01-23)
Category:
answer to question

We are grateful to Reviewer 2 for their thoughtful remarks and for finding our argument convincing.

Concerning the fraction x_a of the dark matter in axions created by the decay of the axion domain walls, it is true that our choice of x_a = 0.5 was rather simplistic. We did note that considerations of structure formation might impose a more stringent constraint. However we agree that taking x_a = 1 is most conservative. Accordingly we have done this and find our bound to now be reduced by a factor of ~2 to 33 meV. Our Figures 2 and 3 have been correspondingly redone with x_a = 1 and appropriate comments have been added in Section 3.3.

We had also missed citing a recent eprint by Notari et al (2211.03799) who carefully calculate the upper bound on the axion mass from considerations of structure formation to be 240 meV. This means that the axion window is reduced to 33-240 meV for DFSZ axions in the post-inflationary scenario. We have now said so in our Conclusions.

---

## Round 3 · Referee Report · Anonymous (Referee 2) · 2023-3-5

Report

The authors have addressed the point that I commented on, and based on my assessment, I can recommend that the paper be published.
  • validity: -
  • significance: -
  • originality: -
  • clarity: -
  • formatting: -
  • grammar: -

Author:  Subir Sarkar  on 2023-03-30  [id 3527]

(in reply to Report 1 on 2023-03-05)

We thank the Referee for acknowledging that we have addressed the point raised and for recommending publication.

---

## Round 3 · Referee Report · Anonymous (Referee 1) · 2023-3-15

Report

I think that the changes performed by the authors go in the right directions. However, I feel that they do not completely respond to the comments I made.

My point about the suppression of the explicit breaking is that the dimensionless coefficient could be extremely small. Indeed, if for instance the effect comes from non-perturbative semi-classical gravitational solutions with action $S$, then the coefficient is $g~\sim\exp(-S)$ which can be much smaller than $10^{-2}$, in which case Figure 1 would look different. If the coupling $g'$ that enters into $S$ is small, then $S$ would be enhanced by inverse powers of $g'$, e.g. as $S\sim 1/g'^2$, and $g$ can be even $e^{-O(100)}$. What would be the lower bound on the allowed dimension of the operator that breaks PQ in this case?

I believe that the problem I mentioned about the `relaxation’ of the biased initial conditions during the evolution of the strings should be discussed in the text. It is a crucial issue that might prevent the bias solution of the domain wall problem to work, even if biased initial conditions were found. I think it would be more useful if the authors mentioned it at the start, so as to allow a reader to be aware of it. This would avoid a reader trying to construct a model with biased initial conditions that might not solve the domain wall problem in the first place, or to think of a possible way out.

The study of the dark matter abundance from domain wall decay has been carried out in the past by several collaborations. The authors often refer to the more detailed simulations of [55,56,58] to support their result, in particular that their estimate of the gravitational wave production differs of a factor of 5 from [55,56,58]. I would be grateful if the authors explained (at least to me) how the dark matter abundance and their bound on the axion mass ($\sim$60 meV) differs from the previous detailed literature ([55,56,58]) and what is the crucial new insight that leads to the new bound. In particular, do the domain walls reach a regime with a constant domain wall area parameter, as in some of the simulations the authors refer to?

As a minor comment, I believe that requiring min $V = 0$ is not precise, because the cosmological constant is small, but believed to be non-zero.

I feel that the authors should at least comment about the issues above before the paper can be published.
  • validity: -
  • significance: -
  • originality: -
  • clarity: -
  • formatting: -
  • grammar: -

Author:  Subir Sarkar  on 2023-03-30  [id 3526]

(in reply to Report 2 on 2023-03-15)

Warnings issued while processing user-supplied markup:

  • Inconsistency: Markdown and reStructuredText syntaxes are mixed. Markdown will be used.
    Add "#coerce:reST" or "#coerce:plain" as the first line of your text to force reStructuredText or no markup.
    You may also contact the helpdesk if the formatting is incorrect and you are unable to edit your text.

We are happy to comment on the issues raised by the Referee - see below:

The prefactor g can indeed be exponentially small - however such models do not have PQ symmetry breaking after inflation hence the issue is irrelevant (as the Editor-in-Charge also notes). We have kept g explicit in all equations and shown the effect of setting g << 1 in the figures - so the effect on the allowed dimension of the operator can easily be assessed.

Concerning the possible 'relaxation’ of biased initial conditions, we have already noted as the Referee suggests at the start of 'Section 2.2 Bias’ (in footnote 2): "However, if the walls are connected by strings, the biased initial conditions mean that the initial string configuration is energetically disfavoured and would likely relax to an unbiased state by emitting axions before the formation of DWs. In any case we find here that bias does not solve the axion DW problem, hence further discussion of this issue is not warranted in the present paper. We have also added a citation to a eprint by Gonzalez et al, arXiv:2211.06849 that appeared after ours which argues that the bias solution does not work when inflationary correlations on superhorizon scales are taken into account.

Regarding how our bound on the axion mass differs from the previous detailed literature [55,56,58] ([56,57,59] in the current version) here are our comments on these papers:

  • Hiramatsu et al. [56] focus 'On the estimation of gravitational wave spectrum from cosmic domain walls’; while their numerical code has been used in previous discussions of axion cosmology, they do not specifically consider axions.

  • Saikawa [59] too provides ‘A review of gravitational waves from cosmic domain walls’ - not specifically about axions.

  • Kawasaki et al [56] discuss 'Axion dark matter from topological defects’ in detail. To answer the Referee, we do assume (our eq.20) that the domain wall network evolution reaches the regime with a constant domain wall area parameter before the wall network collapses, but Kawasaki et al [56] consider both the scaling solution, as well as a small deviation from scaling that is seen in their detailed numerical simulations (modelled as in their eq.3.14). However they conclude, as shown in their Fig.4, that "that there is no significant difference between the assumption of exact scaling and that of deviation from scaling”. Concerning the dark matter abundance, they say "... axions can be responsible for the cold dark matter in the mass range ... m_a ≈ (10^−4 – 10^−2) eV with a mild tuning of parameters for the models with N_DW > 1”. We find however with no tuning of parameters a lower limit (eq.48) on the axion mass that is significantly higher: 3.3x10^−2 eV. As stated in the caption of our Fig.3: "If δ is fine-tuned to be 10^−2, then μ can increase to 10^−8 thus allowing the grey shaded region, i.e. lighter axions down to m_a ∼ 10−2 eV”. Kawasaki et al [56] show their constraints on m_a in their Fig.9 for N_DW = 6 and taking δ to be 10^−3 (=> m_a > 6x10^−3 eV), 10^−5 (=> m_a > 1.6x10^−3 eV) or 10^−8 (=> m_a > 8x10^−4 eV). Using the scaling with δ, these translate into a lower bound of m_a > 4x10−2 eV for δ=1 (taking N_DW=6), to be compared with our bound of m_a > 3.3x10^−2 eV (taking N_DW=2). Another difference in our assumptions is that Kawasaki et al [56] interpret the neutron EDM bound to imply a more restrictive limit on the QCD theta parameter of 7x10^-12 (their eq.4.9), whereas we adopt the, more standard, limit of 10^-10 (our eq.10). Again since our results come from analytic solutions to the governing equations with clearly stated assumptions and choices of parameters (whose scaling is explicitly shown in eq.45) it can be seen that for the same adopted limit, the Kawaski et al [56] lower bound on m_a would change to m_a > 3x10−2 eV (δ=1, N_DW=6). This is close enough to what we find - with a transparent, analytic treatment rather than relying on detailed numerical solutions. Note that some of these were done in previous work (Hiramatsu et al., Phys. Rev. D85 (2012) 105020) and have been improved on in subsequent work by the same authors e.g. Hiramatsu et al. [56]. So whereas the validity of the numerical simulations is established by our analytic result, there may be some uncertainty in the precise bound.

Concerning requiring min V=0, astronomers indeed say that the cosmological constant is not quite zero. However the value they obtain from interpreting observational data in the framework of the assumed maximally symmetric Friedmann-Lemaitre-Robertson-Walker model is \Lambda ~ H_0^2 ~ (10^−42 GeV)^2. This corresponds to an energy density of \Lambda/8\pi G_N ~ (10^-12 GeV)^4 … which is pretty much zero compared to the energy scales being considered here!

We hope our comments above (and in the text) satisfactorily address all the issues raised.

---

## Round 4 · Author Response

Concerning the further points raised by one of the Referees, we are happy that the EIC agrees that it is not necessary to further discuss the possibility that the prefactor g can be exponentially suppressed - as such models will not exhibit post-inflationary PQ symmetry breaking.

  1. We have responded in detail to this Referee's question about the relationship between our results to refs.[55,57,59]. We believe we have provided a transparent, analytic understanding of the numerical results which is of value.

  2. We had indeed misunderstood the question "In the case H_{inf} << f_{PQ}: why wouldn’t this automatically be the pre-inflationary axion scenario?” We agree that H_{inf} is the relevant parameter to compare with f_{PQ} - not V^{1/4}; for H_inf << f_PQ, the PQ symmetry is broken before inflation starts and in this sense this is the pre-inflationary scenario. What we are saying is that despite the fact that there are quantum fluctuations of order H_{inf}, those are too small to ever move the field far from its initial value and therefore will neither serve to generate a bias nor to cross from one vacuum to the other. This was pointed out by Linde & Lyth, Phys. Lett.B 246 (1990) 353 who showed that quantum fluctuations of the axion field during inflation can lead to formation of exponentially big axionic domain walls. We have now cited this paper and clarified the point.

We hope this addresses all the outstanding issues.

---

## Round 4 · List of Changes

1) Added a statement in Footnote 2 (p.7): "Moreover recent lattice simulations show that taking super-horizon inflationary correlations into account can undermine the bias mechanism [39]".

[39] D. Gonzalez, N. Kitajima, F. Takahashi and W. Yin, Stability of domain wall network with initial inflationary fluctuations, and its implications for cosmic birefringence (2022), 2211.06849.

2) Clarified that our limit (48) on f_{PQ} and m_a is consistent with previous numerical results (p.12): "Our bound is consistent with previous numerical work [42, 58]."

3) Added clarification concerning the pre-inflationary axion scenario (p.7): "To generate a bias we must wait until the mean value of the field accumulates around the vacuum states. This can take a long time during which the causally connected patches at the time of PQ symmetry breaking inflate massively and any domain walls that form are exponentially large [52]. Hence this is the same as pre-inflationary PQ symmetry breaking."

[52] A. D. Linde and D. H. Lyth, Axionic domain wall production during inflation, Phys. Lett. B 246, 353 (1990), doi:10.1016/0370-2693(90)90613-B.

---

## Editorial Decision

published